# Aluminum-Doped Zinc Oxide Improved by Silver Nanowires for Flexible, Semitransparent and Conductive Electrodes on Textile with High Temperature Stability

**DOI:** 10.3390/ma16113961

**Published:** 2023-05-25

**Authors:** Maximilian Lutz Hupfer, Annett Gawlik, Jan Dellith, Jonathan Plentz

**Affiliations:** Leibniz Institute of Photonic Technology (IPHT), Albert-Einstein-Str. 9, 07745 Jena, Germany; maximilian.hupfer@leibniz-ipht.de (M.L.H.); jan.dellith@leibniz-ipht.de (J.D.)

**Keywords:** transparent flexible electrode, smart textile, electronic textile

## Abstract

In order to facilitate the design freedom for the implementation of textile-integrated electronics, we seek flexible transparent conductive electrodes (TCEs) that can withstand not only the mechanical stresses encountered during use but also the thermal stresses of post-treatment. The transparent conductive oxides (TCO) typically used for this purpose are rigid in comparison to the fibers or textiles they are intended to coat. In this paper, a TCO, specifically aluminum-doped zinc oxide (Al:ZnO), is combined with an underlying layer of silver nanowires (Ag-NW). This combination brings together the advantages of a closed, conductive Al:ZnO layer and a flexible Ag-NW layer, forming a TCE. The result is a transparency of 20–25% (within the 400–800 nm range) and a sheet resistance of 10 Ω/sq that remains almost unchanged, even after post-treatment at 180 °C.

## 1. Introduction

The ongoing advancement in materials science is enabling increasingly comprehensive integration of electronic components into textiles [1,2,3], which in turn facilitates mobile energy conversion [4], storage and sensor technology close to the user’s body. These electronic components are based on coated fibers or textiles, whose electrical conductivity must be maintained under chemical, mechanical and thermal stresses during further processing and application. To make conventional natural materials or polymers conductive, metals [5,6], (semi)conductive polymers [7,8] and carbon composites [9] are applied via coating methods such as evaporation [10], electroplating [11,12], printing [13,14] or dipping [15,16]. Depending on the electronic components to be integrated, materials are required that are both transparent to visible light and—depending on the layer structure—electrically conductive, enabling a closed carrier layer. The integration of these transparent conductive electrodes (TCE) as invisible conductors allows optoelectronic components, such as solar cells, to function [17] and provides freedom of design in clothing. Transparent conductive oxides (TCO; I:Sn2O, F:Sn2O, Al:ZnO), commonly used for this purpose on non-flexible substrates, exhibit high conductivity and transparency for closed layers, but are rigid. Although their flexibility can be enhanced by applying them in thin layers and in combination with adhesive layers [18], they lag behind polymers or natural fibers used for flexible substrates in terms of thermal expansion and elasticity. Consequently, in addition to the mechanical stress during use, thermal stress from subsequent processes can cause cracks, which increase the sheet resistance of the TCO layer. In addition to the drawback that TCOs tend to crack on flexible substrates, thus reducing the film resistance, the production of TCOs is limited by the materials that can be used for flexible substrates. For instance, gas phase-deposited TCOs must be produced below the temperature required for optimum film quality due to the low melting temperature of most fiber materials (e.g., 200 °C for polymers). This further increases the film resistance [18,19].

Therefore, for use as a TCE on flexible fibers, a material that can be processed at low temperatures with low sheet resistance and high temperature stability is required. In this regard, metal nanowires (NWs) are a promising alternative for replacing TCOs with TCEs [20]. For example, silver nanowires can be wet-chemically deposited onto textiles at low temperatures [21] and span fibers depending on their length, thus establishing a conductive as well as flexible network [22]. This allows the realization of a non-area-covering electrode with a transparency of 61% and a conductivity of 7.3 Ω/sq [21]. Ag-NWs are superior to other transparent stretchable conductors made from carbon-based nanomaterials (CNTs and graphene) or polymers due to their excellent electrical conductivity and high aspect ratio [23,24].

With the aim of a closed TCE, wet-chemically deposited ZnO nanoparticles and Ag-NW were combined on graphene and a sheet resistance of 1.5 Ω/sq was achieved [25]. Nevertheless, for these NW networks, again the surface coverage determines the balance between electrical conductivity and transparency of the layer [22], so that a closed TCE, such as can be achieved with TCOs and is required for solar cells, is not possible. In this respect, it has been shown on foils that combining Ag-NW with Al:ZnO retains the advantages of both materials to achieve a semi-transparent, conductive, flexible and low temperature processable TCE [26,27,28,29]. Thus, the resulting foil-based TCE has a high transparency of 64–90% with a low sheet resistance of 6–20 Ω/sq [27,28,29,30]. By combining silver nanowires and Al:ZnO, the sheet resistance could be almost maintained (<10% loss) during mechanical bending with a bending radius of 2.5 mm [28]. Even when subjected to mechanical bending with a bending radius of 3 mm and thermal stress of up to 250 °C, the TCE of silver nanowires and Al:ZnO enabled the sheet resistance to be nearly maintained [30]. However, compared to a foil, the textile with its woven threads has a more complex 3D structure. Therefore, with regard to a textile-integrated closed TCE, it is important to determine for the material combination how the surface coverage affects the optoelectronic properties due to possible cracking under mechanical and thermal stress.

Therefore, in the present work, and for the first time, a Ag-NW network supplemented by an Al:ZnO layer is integrated onto a textile. The structural and optoelectronic properties of this textile-based flexible TCE are measured before and after mechanical and thermal stress. To characterize the material properties on the textile, the conductivity and sheet resistance are determined by two- and four-point methods. The surface coverage and condition of the substrates and possible crack formation in the conductive layer were investigated by optical and electron microscopy.

## 2. Materials and Methods

### 2.1. Cleaning Procedure

The polyamide-based textile support substrates provided by ITP GmbH were uniformly cut to a size of 2.5 × 2.5 cm. Each substrate was soaked for 15 min in 2 vol% Hellmanex II^®^ (in water), acetone and isopropanol to remove organic residues. Ultrasonics were not used as this would lead to fragmentation of the cut fabrics. After the solvent cleaning, the substrates were treated with oxygen plasma for 10 min to improve the adhesion of the coating to the polymer.

### 2.2. Material Deposition

For deposition of the silver nanowires (Ag-NW; width = 40 ± 5 nm, length = 35 ± 5 µm, concentration = 5 mg/mL in EtOH from SigmaAldrich, St. Louis, MO, USA), the substrates were immersed in a diluted dispersion (concentration = 2 mg/mL in EtOH). For each Ag-NW coating cycle, the textile substrates were stirred in the dispersion for 30 s and then dried in an oven at 40 °C for 5 min. This coating cycle was repeated up to 7 times. For the coating with aluminum-doped zinc oxide (Al:ZnO), the samples were cleaned again using a 10 min oxygen plasma treatment, and then a film of 500 nm thickness was deposited by atomic layer deposition (ALD, Beneq TFS200). The ALD process is a gas-phase deposition process and was preferred because all filaments and threads are completely coated with the Al:ZnO material. The deposition rate used was 0.25 nm/cycle at a working pressure of 0.5 mbar. A pulse duration of 200 ms and an operating temperature of 180 °C (175, 200, 225 and 250° for determining the optimum coating temperature on glass) was applied. The gases trimethylaluminum and diethylzinc were used as precursors with a concentration of 1:20, and the purging gas, N_2_, was used to remove the unreacted precursors.

### 2.3. Mechanical and Thermal Test

For the bending tests, the substrates were placed on surfaces with different radii and the resistance was determined by 2-point measurements during bending. A maximum bending of 210° used in this study, and the bending and relaxation process was repeated for up to 200 cycles. Adhesion was measured using a strip test. For this, an adhesive tape (Magic Scotch Tape) was applied over the sample and pressed down. The tape was then removed in a smooth, even motion. To test the thermal stress, the substrates were heated in an oven from room temperature to the loading temperature. After heat treatment, the substrates were removed from the oven and cooled at room temperature.

### 2.4. Electrical Characterization

Sheet resistance was determined using a custom-made 4-point probe measurement in a linear configuration. The mean value and the standard deviation of the sheet resistance were determined per coating cycle from 4 samples with 4 current values per sample. The minimum sheet resistance of the Ag-NW mesh as a function of coating cycles was calculated by fitting a two-phase exponential decay function to the experimental data using Origin Lab 8.5. The mean value and standard deviation of the line resistance were determined from 4 samples by measuring at opposite ends of the substrates (length = 2 cm) before, during and after bending using a 2-point probe measurement using a multimeter.

### 2.5. Optical Characterization

A UV-vis spectrometer (Varian: Cary 5000) was used in transmission mode to measure both the blank and deposited textiles. Transmission spectra were measured in the range of 300 to 800 nm with a step size of 1 nm.

### 2.6. Surface and Chemical Characterization

The quality of the coating was assessed using light microscopy and scanning electron microscopy (SEM). Light microscopic images were measured at 20× magnification in incident light in the bright field (AxioImager.Z1, Carl-Zeiss, Oberkochen, Germany). The majority of the SEM images were obtained using a LYRA XMU (Tescan, Brno, Czech Republic) and an Everhart-Thornley Detector for secondary electrons. The electron energy was set to 5 keV in all cases. The images were measured at 1 kx and 5 kx magnification. Furthermore, X-ray mappings were captured using a silicon drift detector, Xflash 5030 (Bruker Nano GmbH, Berlin, Germany), to illustrate the element homogeneity. In this case, the energy of the exciting electrons was set to 15 keV and energy windows were defined in the X-ray spectra for the elements of interest.

## 3. Results and Discussion

The polyamide-based textile was coated with Ag-NW and the film quality of the deposition was evaluated electrically via the four-point method and optically by light microscopy. The resulting sheet resistance (R_sh_) is depicted as a function of deposition cycles in Figure 1. After the first cycle, no electrical conductivity of the layer could be detected and the images from the optical microscope in Figure 2 show that the nanowires could not form a closed network. For the subsequent depositions, it was determined that R_sh_ systematically decreases from 4.3 ± 2.2 MΩ/sq to 24.7 ± 4.5 Ω/sq within seven cycles. The progression of R_sh_ follows a two-phase exponential decay function, tending towards a minimum of 19.3 ± 0.5 Ω/sq (see Figure 1). Correspondingly, microscope images show a closed and condensed network of Ag-NW that spans fibers of the textile (Figure 2b,c). Subsequently, agglomerates are formed whose number and size increase with each successive immersion steps; see Figure 2. The high standard deviation is due to the poor adhesion of the Ag-NW and the manual nature of its coating. This leads to an uneven distribution, especially at low coating cycles, resulting in a large variation in sheet resistance.

Subsequently, the effect of the Ag-NW network on the conductivity and surface properties of the textiles was investigated. These textiles were further coated with 500 nm aluminum-doped zinc oxide (Al:ZnO) by atomic layer deposition (ALD). Thus, due to the low thermal stability of the polyamide, the Al:ZnO could only be deposited at a reduced temperature of 180 °C, which degrades Rsh compared to the optimal coating temperature of 225 °C (see Appendix A for 400 nm on glass). By combining Ag-NW and Al:ZnO, Rsh could be improved compared to the individual coatings. Therefore, using Al:ZnO (39.7 ± 4.2 Ω/sq), a reduction from 576.8 ± 302.5 to 21.1 ± 2.6 Ω/sq was achieved for the textile coated with four Ag-NW immersion cycles and from 24.7 ± 4.5 to 10.2 ± 2.2 Ω/sq for seven cycles, as seen in Table 1 and Figure 1. The scanning electron microscope (SEM) and energy dispersive X-ray spectroscopy (EDX) mappings in Figure 3 show that the Al:ZnO is homogeneously distributed on the Ag-NW coated textiles, although Al:ZnO tends to form nanocrystals.

For use as a TCE, a semi-transparent fabric was coated with the Ag-NW and Al:ZnO combination, and its transmission was measured using UV-vis transmission spectroscopy. As shown in Figure 4, the uncoated fabric exhibited a transparency of about 45% over the broad spectrum of visible light. The Al:ZnO coating, without Ag-NW, reduced the transparency to approximately 40%. With each Ag-NW deposition cycle, transmission decreased until a transmission of 20–25% (400–800 nm) was achieved after seven cycles. In combination with Al:ZnO, the transmission of the fabric decreased to 5–15% (400–800 nm) at seven deposition cycles. Transmission decreases monoexponentially with the number of coating cycles, as shown in Appendix A. This is due to the enhanced light scattering of the combined layer of Ag-NW and Al:ZnO, causing the overall layer to lose transmission compared to the individual layers.

To verify the mechanical strength and flexibility, the line resistance (R_L_) of the coated textiles was measured as a function of the bending angle φ (Figure 5a) using the two-point method. It was found that for both the pure Al:ZnO layer and the material combinations with Ag-NW, the preservation of R_L_ can be guaranteed when the textile is bent. R_L_ increases for Al:ZnO at φ = 210° from 49.2 ± 7.3 to 51.0 ± 7.7 Ω, but only marginally within the standard deviation. Similarly, for the Ag-NW-supported Al:ZnO films, R_L_ increases only slightly from 20.8 ± 3.1 to 22.6 ± 3.8 Ω and from 10.0 ± 0.6 to 11.2 ± 1.4 Ω for four and seven deposition cycles, respectively.

The process of completely bending from 0 to 210° was repeated an additional 200 times, with R_L_ measured throughout the process. Figure 5b shows that for all three layers shown, R_L_ increases abruptly within the first ten bending cycles. With the subsequent cycles, R_L_ continues to increase, albeit at a slower rate. The images of the scanning electron microscope in Figure 6 demonstrate that after 200 bending cycles, the surface of the coated textiles is almost undamaged and only isolated cracks form along the fiber direction.

To assess the mechanical stability of t-TCE, an adhesion test was performed. Figure 7a shows the change in sheet resistance, denoted as dR/R_0_ = (R-R_0_)/R_0_, following the removal of an adhesive strip. Without Ag-NW support, dR/R_0_ = 1200 ± 160%, while with Ag-NW it improves to 540 ± 220% and 100 ± 30% for four and seven coating cycles, respectively. The high standard deviation results from manual adhesion tests. Figure 7b shows the t-TEC surface after the adhesion test, with the black areas representing adhesive residues. These residues indicate that the layer isn’t peeling off. Additionally, cracks are visible, but the Al:ZnO layer remains around the fiber. Although these cracks reduce the layer resistance, the Ag-NW lattice compensates for the loss of resistance, ensuring stability and performance.

In terms of thermal stress, the individual components of the conductive textile’s material composition have different thermal expansion coefficients α (PA 6: 80 K^−1^, Ag: 19 K^−1^, Al:ZnO 2.8 K^−1^), leading to potential cracking in the most rigid layer (Al:ZnO) as the temperature rises. Therefore, to examine the thermal stability, the substrates were heated below the melting temperature of PA in the oven at 60, 120 and 180 °C and the resulting dR/R_0_ was plotted in Figure 8. It was found that for all three substrate coatings presented, Rsh increases only slightly (dR/R_0_ ≤ 20%) due to heating at temperatures ≤120 °C. While at temperatures >120 °C, R_sh_ of Al:ZnO-coated textiles deteriorates only slightly with the support of Ag-NW (dR/R_0_ ≤ 25%), whereas there is a sharp increase without the Ag-NW (dR/R_0_ = 112.9 ± 11.4%). This change in sheet resistance is accompanied by different patterns of crack formation. As revealed by SEM (see Figure 9), purely Al:ZnO-coated substrates exhibit cracks transverse to the fiber direction, while Ag-NW supported substrates display occasional cracks running longitudinally to the fiber direction. It should be noted, however, that such longitudinal cracks are more easily observed in the clearly spanned surfaces of the Ag-NW supported substrates. It is assumed that even with the pure Al:ZnO coating, the thermal stress between the fibers connected by the coating is discharged in longitudinal cracks. Thus, due to the combination of both transverse and longitudinal cracks, R_sh_ increases more without Ag-NW.

To account for the differences in resistance change under mechanical and thermal stress, the respective length change after bending $dL_{B}=\frac{\pi \varphi }{180{}^{\circ}}$ and thermal expansion $dL_{T}\approx \alpha L_{0}∆T$ were calculated. With a bending angle of φ = 210° and a filament length of 2.5 cm, it was possible to determine a $dL_{B}$ between PA and Al:ZnO of 24 µm.

This simple calculation of $dL_{B}$ did not take into account that in the textile the woven structure mitigates the mechanical stress on the individual filaments (Table 2).

As a result, it is assumed that the actual $dL_{B}$ values are even lower. However, this is countered by the thermal expansion of the complete textile, which, considering different α values, can lead to dL_T_ = 308 µm. Therefore, the stress resulting from mechanical bending in the examined area is always less than the stress due to thermal expansion.

## 4. Conclusions

A semi-transparent conductive electrode integrated into textiles (t-TCE), comprising a combination of silver nanowires (Ag-NW) and Al:ZnO, was examined for its mechanical and thermal stability, as well as transparency on a polyamide-based textile. This combines the advantages of a closed conductive (Al:ZnO) and a flexible (Ag-NW) layer in a TCE, resulting in a transmission of 20–25% (400–800 nm) and a sheet resistance of 10 Ω/sq, which remained nearly unchanged despite post-treatment at 180 °C.

The optoelectronic properties of the t-TCE under mechanical and thermal stress are determined by the surface coverage of the Ag-NW network. The transmission and electrical conductivity of the t-TCE demonstrate an inverse dependence in this context. The sheet resistance (R_sh_) of the layer was measured as a function of the dip-coating cycles of the Ag-NW and a minimum R_sh_ = 19 Ω/sq was determined for a fully closed layer. Semi-transparent designed R_sh_ of the Ag-NW network, combined with 500 nm Al:ZnO decreased from an average of 577 and 25 Ω/sq to 21 and 10 Ω/sq for 4 and 7 dip cycles, respectively. Compared to the uncoated textile, transmission was reduced from 40–45% to 5–20% and 2–15% for four and seven immersion cycles of Ag-NW and Al:ZnO, respectively. The t-TCE’s flexibility was confirmed through mechanical bending tests. It was found that even at 210° bending with 200 repetitions, only minor crack formation occurs in the conductive layer, resulting in only a marginal increase in conductive resistance R_L_. In contrast, mechanical adhesion tests show that clear cracks form locally when an adhesive strip is pulled off, but the layer remains intact. The silver nanowire layer counteracts the loss of sheet resistance associated with the cracks. Consequently, the change in sheet resistance is 1200% without Ag-NW and only 100% with Ag-NW. Compared to mechanical bending, the textile and t-TCE expand more under thermal stress. Therefore, due to the different expansion coefficient, both longitudinal and transverse cracks occur in the Al:ZnO layer, which dramatically increase R_sh_ at temperatures above 120 °C. With the support of Ag-NW, the cracking in the Al:ZnO could be countered, so that R_sh_ remains nearly unchanged at up to 180 °C. To further enhance the conductivity and interconnectivity of the textile-based TCE presented here, the integration of microwires and metal yarns could be considered [31,32].

## Figures and Tables

**Figure 1 materials-16-03961-f001:**
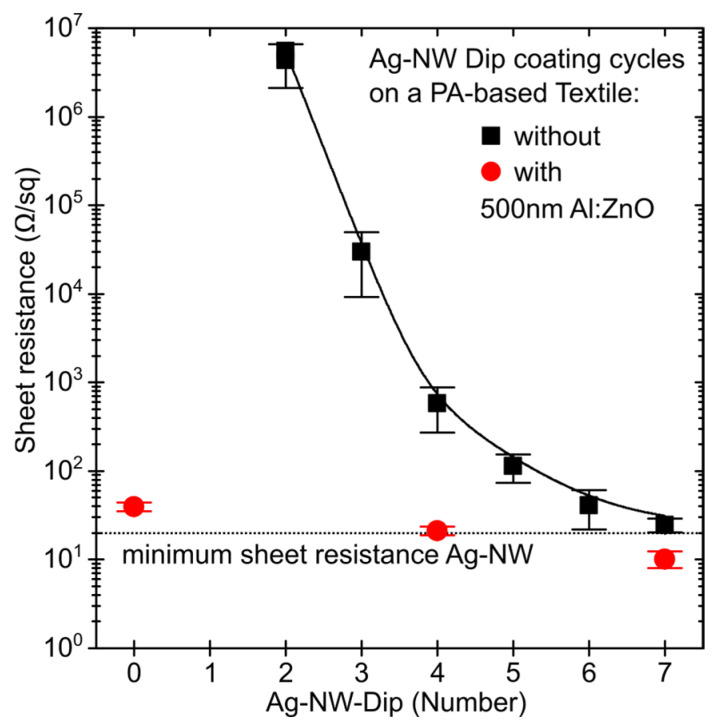
Sheet resistance of a polyamide-based calendered textile as a function of the immersion in a Ag-NW dispersion (2 mg/mL in EtOH, 30 s per dip) with (red) or without (black) an additional Al:ZnO (500 nm @180 °C) coating. The black dotted line indicates the minimum sheet resistance calculated by two-phase exponential decay function fit on the experimental data.

**Figure 2 materials-16-03961-f002:**
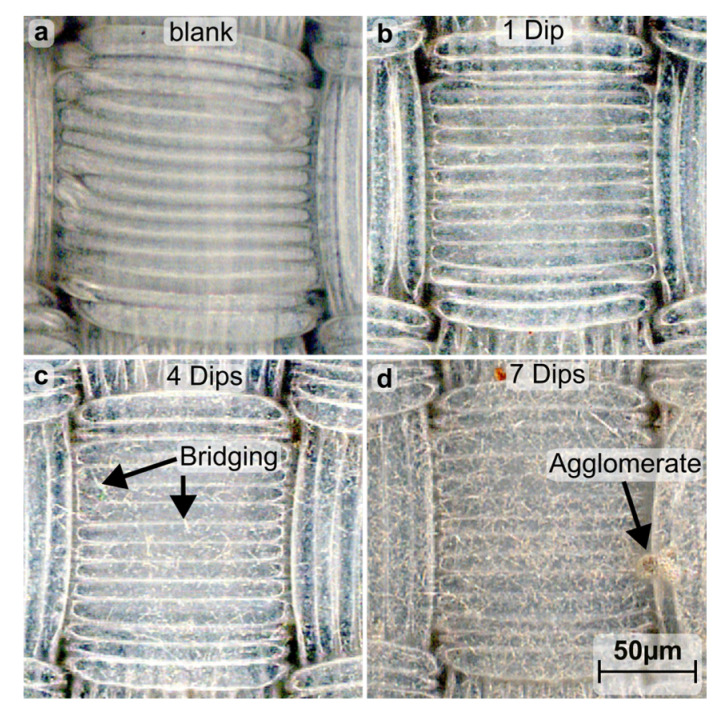
Microscopy image of a polyamide-based calendered textile before (**a**) and after 1 (**b**), 4 (**c**) and 7 (**d**) times immersion in a Ag-NW dispersion (2 mg/mL in EtOH, 30 s per dip) without Al:ZnO.

**Figure 3 materials-16-03961-f003:**
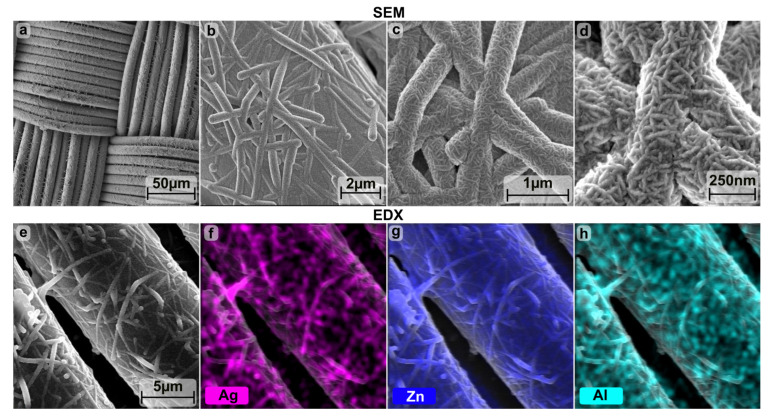
SEM (**a**–**e**) and EDX (**f**–**h**) image of polyamide-based calendered textile coated with Al:ZnO and combined with 4 dips of Ag-NW.

**Figure 4 materials-16-03961-f004:**
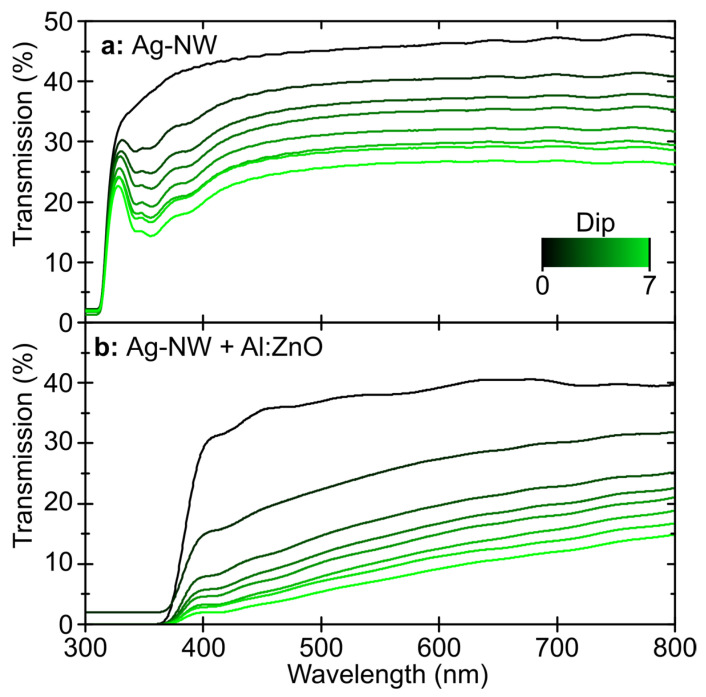
UV-vis transmission spectra of a polyamide-based semi-transparent textile coated with Ag-NW (0–7 Dips) and combined with Al:ZnO.

**Figure 5 materials-16-03961-f005:**
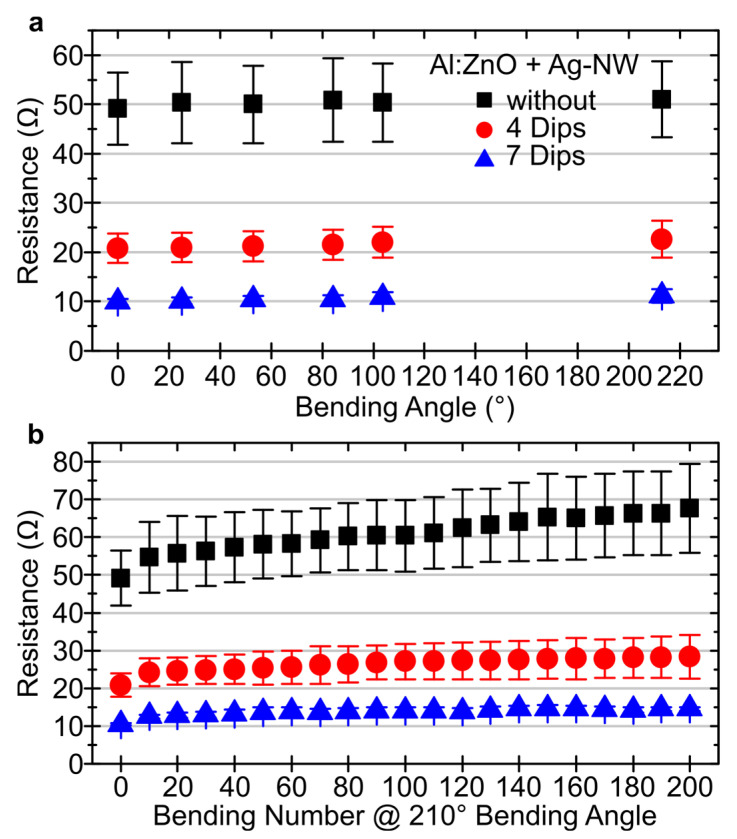
Line resistance in dependence on the (**a**) bending angle and (**b**) bending number (@210°) of a polyamide-based calendered textile coated with Al:ZnO and combined with 0 (black), 4 (red) and 7 (blue) dips of Ag-NW.

**Figure 6 materials-16-03961-f006:**
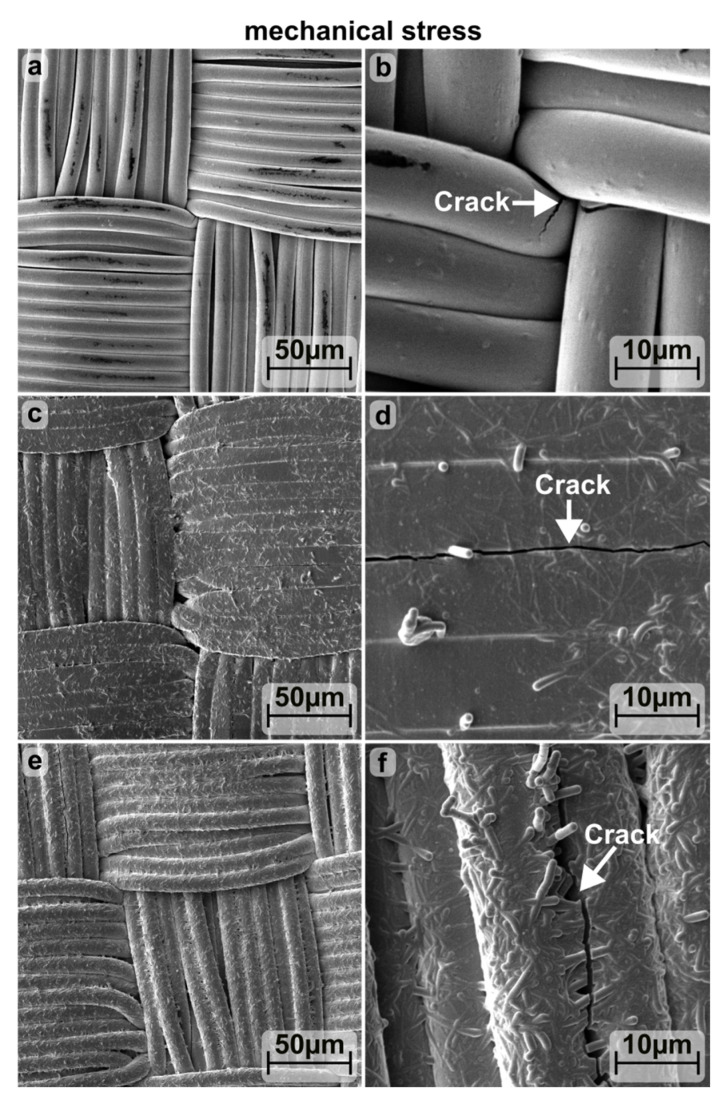
SEM image of polyamide-based calendered textile coated with Al:ZnO, bended 200 times and combined with 0 (**a**,**b**), 4 (**c**,**d**) and 7 (**e**,**f**) dips of Ag-NW.

**Figure 7 materials-16-03961-f007:**
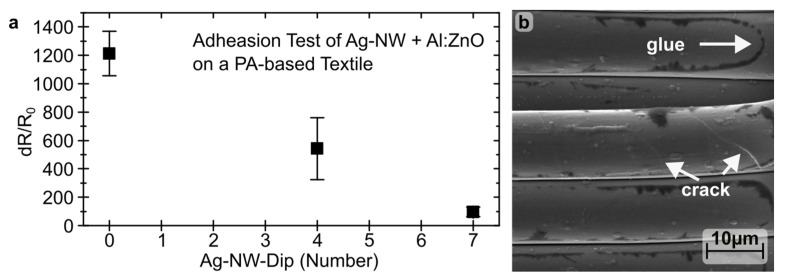
Change in surface resistivity (**a**) and SEM image (**b**) of the combination of Al:ZnO and Ag-NW on a polyamide-based calendered textile after removal of an adhesive tape.

**Figure 8 materials-16-03961-f008:**
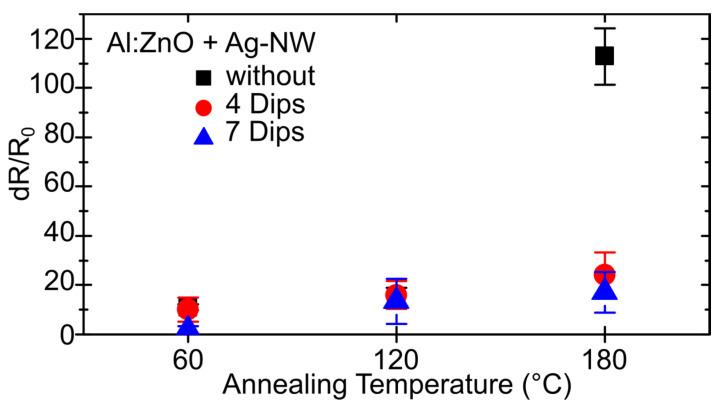
Sheet resistance change of a polyamide-based calendered textile coated with Al:ZnO and combined with 0 (black), 4 (red) and 7 (blue) dips of Ag-NW after thermal treatment of 60, 120 and 180 °C.

**Figure 9 materials-16-03961-f009:**
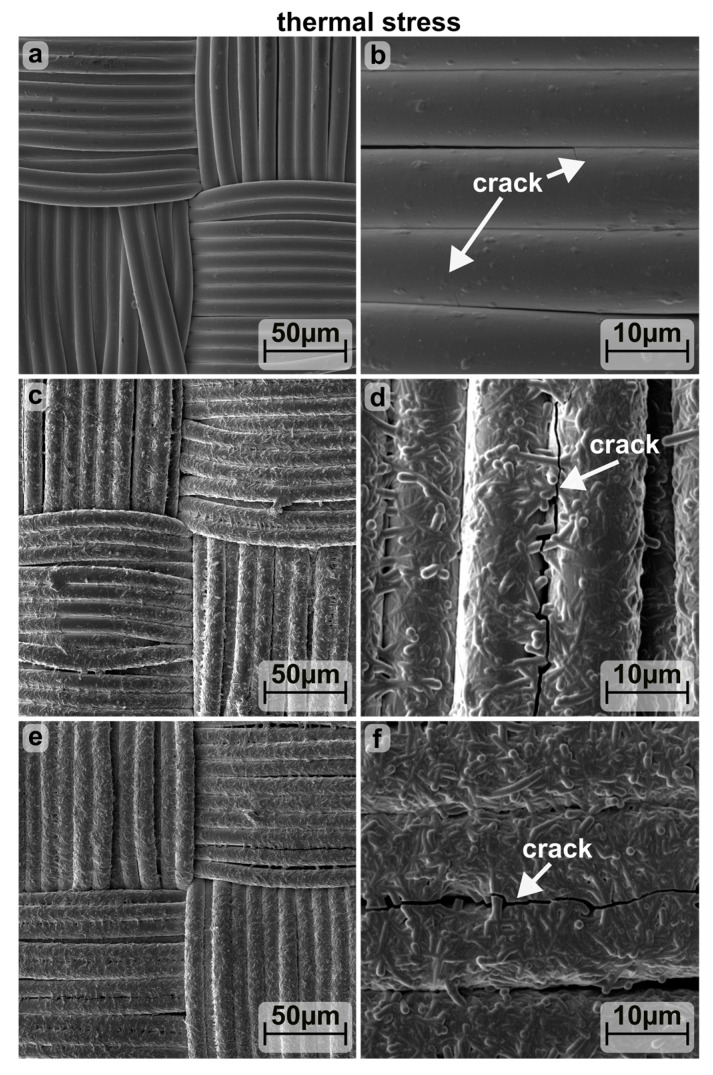
SEM image of a 180 °C annealed polyamide-based calendered textile coated with Al:ZnO and combined with 0 (**a**,**b**), 4 (**c**,**d**) and 7 (**e**,**f**) dips of Ag-NW.

**Table 1 materials-16-03961-t001:** Sheet resistance of a polyamide-based calendered textile after 0, 4 and 7 dip coating cycles of a Ag-NW dispersion (2 mg/mL in EtOH, 30 s per dip) with or without an additional Al:ZnO (500 nm @180 °C) coating.

Number of Dip Coating Cycles	Sheet Resistance of Ag-NW Coating (Ω/sq)	Sheet Resistance of Ag-NW + Al:ZnO Coating (Ω/sq)
0	∞	39.7 ± 4.2
4	576 ± 302	21.1 ± 2.6
7	24.7 ± 4.5	10.2 ± 2.2
∞	19.3 ± 0.5	/

**Table 2 materials-16-03961-t002:** Difference (dL = dL_PA_-dL_Al:ZnO_) in length change of polyamide (dL_PA_) and Al:ZnO (dL_Al:ZnO_) due to bending at an angle φ = 210° and thermal expansion with the thermal expansion coefficient α = 80 1/K (PA 6) and 2,9 1/K (Al:ZnO) with an initial length L_0_ = 2.5 cm.

	Condition	dL_PA_ (µm)	dL_Al:ZnO_ (µm)	dL (µm)
Beding:	φ = 210°	26.0	1.9	24.1
Thermal:	T = 60 °C	80.0	2.9	77.1
	T = 120 °C	200	7.3	192
	T = 180 °C	320	11.6	308

## Data Availability

The data is stored on the servers of the Leibniz Institute of Photonic Technologies and is available on request at any time.

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
