# Peer review of "Aluminum-Doped Zinc Oxide Improved by Silver Nanowires for Flexible, Semitransparent and Conductive Electrodes on Textile with High Temperature Stability"

_materials, 2023, doi:10.3390/ma16113961_

Round 1

Reviewer 1 Report

Reviewers comments of manuscript, materials-2385659, Aluminum-doped Zinc oxide improved by silver nanowires for flexible, semitransparent and conductive electrodes on textile with high temperature stability

The manuscript deals with the study of aluminum-doped zinc oxide combined with silver nanowires for flexible, semitransparent and conductive electrodes on textile with high temperature stability, even importantly their potential application results was presented in details in this MS. The quality of this MS probably validates its publication in Materials, but it is not acceptable for publication in its present form. However, major revisions and corresponding clarify a couple of issues need to been done by the authors, as follow:

1. In introduction part, the authors should illustrate the specific reason or application advantage of aluminum-doped zinc oxide combined with silver nanowires in comparison with other type methods. Meanwhile, the author also should refer to the latest progress and applications of transparent conductive electrodes (TCE) of silver nanowires materials. Therefore, it is quite necessary to add the related works or statements, and then elicit the research significance of aluminum-doped zinc oxide. To some extent, the silver nanowires (Ag-NW) are conventional, and also lacks the novelty, so the authors should give the main reasons and consideration for the experimental design and explanation both in introduction and discussion parts.

2. According to this MS, the authors present a large of experimental data, including the measurements of flexible, semitransparent and conductive electrodes. Therefore, it is possible to conclude or understand the corresponding relations for each other (such as flexible, semitransparent and conductive properties), so I strongly suggest the author should build a simple model, and supplement the related statement for the final explanation.

3. I have just listed a few but not all plotting and typing mistakes below:

Figure 1:

I suggest the authors should redraw the images, and export the images with high-resolution.

Table 1:

There is some wrong with table, I suggest the authors should re-design them separately.

Figure 2:

I strongly suggest the authors should remark all the images with the larger symbols and serial numbers.

Figure 4:

I suggest the authors should magnify the scale and caption in each image.

Figure 5:

I suggest the authors should redraw all the images, similar as Figure 1, respectively.

Figure 7:

I suggest the authors should redraw the image, similar as Figure 1.

Conclusions part:

I suggest that the author should supplement the 2~3 items of conclusion sentences for quick information and reading.

References:

The journal abbreviation and typesetting in references should be done strictly according to the demand and rule of Materials. i.e. Ref.[1], Ref.[5], Ref.[6], Ref.[8], Ref.[9], Ref.[10], Ref.[11], Ref.[15], Ref.[20], Ref.[22], Ref.[29], Ref.[31], etc.

In addition, the authors should add some related references for microwires into MS, such as: https://doi.org/10.1002/adem.201000204, http://dx.doi.org/10.1016/j.matdes.2016.01.090, etc.

English writing:

English writing of this MS could be improved further.

……

English writing:

English writing of this MS could be improved further.

Reviewer 2 Report

This paper reports on the use of silver nanowires to improve aluminum-doped zinc oxide to be used in flexible conductive electrodes for textiles.

Overall, the structure of the work is correct, and the manuscript is easy to read. The figures used help to understand the information contained in the text. Along this paper, the authors made an effort to correctly describe the experimental work regarding the production and characterization of their samples. The manuscript is well organized and clear when analyzing the results.

It is my opinion that this work should be published in Materials with very small minor revisions. I think that only small issues, listed below, prevent this work to be published as is. 

Minor revisions: 

1 – Page 3, line 99:  The sentence “For the maximum bending of 210° used in this publication, the bending and relaxation process was repeated up to 200.” is missing one word in the end. I think that word could be “cycles”, but the authors should know what word is best to complete the sentence. 

2 – Page 4, line 150: The authors present the value 576.8 ± 302.5 Ω/sq. This value has a very high standard deviation. Am I correct in assuming that this high standard deviation is due to the low number of immersion cycles? The standard deviation becomes smaller (and acceptable) when the number of immersion cycles increase. The authors never mention this in the text. 

3 – Page 8, line 225: There is a format problem here. I do not know if this is an issue with the copy I received, but this sentence is not complete and there is a reference to Table 2 that seems to be out of place.

Reviewer 3 Report

This manuscript entitled "Aluminum-doped zinc pxide improvedby silver nanowires foe flexible, semi-transparent and conductive electrodes on textile with high temperature stability" is interesting and well-organized.  It is suitable to publish in Materials after minor revision.  Comments are given as follows:

1. Please provide the adhesion testing between the TCO and Ag-NW. 

Round 2

Reviewer 1 Report

The authors have well done for all of my comments. 

In addition, the English should be improved finally before the publication.

Author Response

Reviewer 1:

Reviewer 1: Comment 1: “In addition, the English should be improved finally before the publication.”

Our Response: We used the time to revise our entire text with a native speaker.

Change in the manuscript: Due to the extent of the revisions, we refer to the revised manuscript.
